# Evaluation of a Novel Dry Powder Surfactant Aerosol Delivery System for Use in Premature Infants Supported with Bubble CPAP

**DOI:** 10.3390/pharmaceutics15102368

**Published:** 2023-09-22

**Authors:** Robert M. DiBlasi, Coral N. Crandall, Rebecca J. Engberg, Kunal Bijlani, Dolena Ledee, Masaki Kajimoto, Frans J. Walther

**Affiliations:** 1Department of Respiratory Care Therapy, Seattle Children’s Hospital, Seattle, WA 98105, USA; 2Center for Respiratory Biology and Therapeutics, Seattle Children’s Research Institute, Seattle, WA 98101, USA; coral.crandall@seattlechildrens.org (C.N.C.); rebecca.engberg@seattlechildrens.org (R.J.E.); masaki.kajimoto@seattlechildrens.org (M.K.); 3Quality and Clinical Effectiveness, Seattle Children’s Hospital, Seattle, WA 98105, USA; 4Center for Clinical and Translational Research, Seattle Children’s Research Institute, Seattle, WA 98101, USA; 5Mechanical Engineering, Zewski Corporation, Magnolia, TX 77354, USA; kunalb@zewski.com; 6Division of Cardiology, Department of Pediatrics, University of Washington, Seattle, WA 98195, USA; dolena.ledee@seattlechildrens.org; 7Department of Pediatrics, David Geffen School of Medicine, University of California Los Angeles, Los Angeles, CA 90095, USA; fjwalther@ucla.edu; 8Lundquist Institute for Biomedical Innovation at Harbor-UCLA Medical Center, Torrance, CA 90502, USA

**Keywords:** dry powder lung surfactant, synthetic lung surfactant, aerosol delivery, preterm infant airway model, bubble CPAP, nebulizer, nasal prongs, humidification, particle size

## Abstract

Aerosolized lung surfactant therapy during nasal continuous positive airway pressure (CPAP) support avoids intubation but is highly complex, with reported poor nebulizer efficiency and low pulmonary deposition. The study objective was to evaluate particle size, operational compatibility, and drug delivery efficiency with various nasal CPAP interfaces and gas humidity levels of a synthetic dry powder (DP) surfactant aerosol delivered by a low-flow aerosol chamber (LFAC) inhaler combined with bubble nasal CPAP (bCPAP). A particle impactor characterized DP surfactant aerosol particle size. Lung pressures and volumes were measured in a preterm infant nasal airway and lung model using LFAC flow injection into the bCPAP system with different nasal prongs. The LFAC was combined with bCPAP and a non-heated passover humidifier. DP surfactant mass deposition within the nasal airway and lung was quantified for different interfaces. Finally, surfactant aerosol therapy was investigated using select interfaces and bCPAP gas humidification by active heating. Surfactant aerosol particle size was 3.68 µm. Lung pressures and volumes were within an acceptable range for lung protection with LFAC actuation and bCPAP. Aerosol delivery of DP surfactant resulted in variable nasal airway (0–20%) and lung (0–40%) deposition. DP lung surfactant aerosols agglomerated in the prongs and nasal airways with significant reductions in lung delivery during active humidification of bCPAP gas. Our findings show high-efficiency delivery of small, synthetic DP surfactant particles without increasing the potential risk for lung injury during concurrent aerosol delivery and bCPAP with passive humidification. Specialized prongs adapted to minimize extrapulmonary aerosol losses and nasal deposition showed the greatest lung deposition. The use of heated, humidified bCPAP gases compromised drug delivery and safety. Safety and efficacy of DP aerosol delivery in preterm infants supported with bCPAP requires more research.

## 1. Introduction

Each year 1 million preterm infants die from respiratory distress syndrome (RDS) in low- and middle-income countries (LMICs) due to a lack of respiratory support devices and surfactant therapy [1]. Standard treatment for preterm infants with surfactant deficiency and RDS in high-resource settings includes timely intratracheal instillation of animal-derived liquid surfactant through an endotracheal tube in combination with mechanical ventilation, which aids in liquid dispersion and displacement of the course liquid from the conducting airways to the acinar regions of the lungs. This process can result in complications such as acute airway obstruction, hypoxemia, hypercarbia, hemodynamic instability, pulmonary hemorrhage, pneumothorax, and non-uniform medication distribution of surfactant in the lungs [2]. Surfactant therapy and mechanical ventilation are usually not an option in many LMICs due to high surfactant costs and the lack of equipment and skilled personnel needed to intubate, operate, and maintain ventilators in preterm infants [3,4]. Nasal continuous positive airway pressure (CPAP) is associated with less need for surfactant therapy and less lung injury and bronchopulmonary dysplasia (BPD) than invasive ventilation [5]. However, CPAP failure rates in LMICs are high despite the wide availability of bubble nasal CPAP (bCPAP) [6,7], and there is a need to support those infants who fail bCPAP with surfactant therapy.

Aerosolized surfactant with small, inhaled particles generated by a nebulizer has been shown to produce more uniformly distributed medication deposition deep into the lungs [8] and fewer adverse effects on blood pressure and cerebral blood flow than standard liquid instillation in ventilated subjects [9]. To bridge a major gap in lung protection and infant mortality, combining the aerosol delivery of surfactant with bCPAP could extend the functional capabilities of noninvasive support and prevent intubation in a large fraction of infants that would otherwise fail bCPAP, require invasive ventilation, or die from surfactant deficiency in LMICs. Findings from clinical trials have shown a reduced need for intubation, but aerosol treatment has not been associated with a decreased risk of death, BPD, or other neonatal morbidities compared to standard therapy. Combining surfactant aerosol with CPAP has been met with significant challenges with reported low nebulizer efficiency, high impactive losses within the CPAP prongs, gas circuits, and upper airways, and low pulmonary deposition [10,11]. While nebulizer efficiency has improved, most nebulizers produce aerosol throughout the respiratory cycle, and a significant amount of the drug is lost during the exhalation phase. This often requires multiple surfactant doses, which is not economical for infants in LMICs.

Dry powder (DP) surfactant aerosol could provide higher pulmonary aerosol concentrations over a shorter dosing period than nebulized liquid surfactant. DP surfactant aerosol combined with an inhaler in-line with a manual resuscitator and ventilator was reported over 40 years ago in preterm infants with RDS by Morley et al. [12,13]. However, if the DP surfactant is not conditioned properly or the particles are too large, this can lead to significant agglomeration and aggregation of the DP aerosols in the airways [14]. More recently, Gninzeko et al. [15] reported a technique using the excipient enhanced growth (EEG) of micrometer-sized particles of an animal-derived DP surfactant in surfactant-deficient rats. Walther et al. [16] applied a synthetic DP surfactant formulation with a simple low-flow aerosolization chamber (LFAC) and bellows bottle for timed inspiratory aerosol medication delivery with bCPAP. They showed improved oxygenation and lung mechanics in preterm lambs.

We conducted pre-clinical in vitro studies with the DP surfactant aerosol delivery system (LFAC) described by Walther et al. [16] to evaluate particle size (mass median aerodynamic diameter, MMAD), operational feasibility, and aerosol delivery efficiency with bCPAP. We hypothesized that there would be no differences in nasal airway deposition or inhaled lung dose between different nasal interfaces and bCPAP system gas humidity levels during concomitant surfactant aerosol therapy and bCPAP in a spontaneously breathing human preterm infant model.

## 2. Materials and Methods

### 2.1. B-YL: Trehalose Synthetic DP Surfactant Formulation

B-YL, a 41-amino acid peptide mimic of surfactant protein B (SP-B) [17], was synthesized using a standard Fmoc protocol, cleaved, purified with reverse-phase HPLC, quantified, and had its mass confirmed. Acorda Therapeutics Inc. (Waltham, MA, USA) used proprietary ARCUS^®^ pulmonary DP technology to formulate the B-YL: Trehalose surfactant by adding 49 weight% of DPPC and 21 weight% of POPG-Na (Avanti Polar Lipids, Alabaster, AL, USA), 25 weight% of the excipient Trehalose (Sigma-Aldrich Co., Saint Louis, MO, USA), 3 weight% of B-YL, and 2 weight% of NaCl (Sigma-Aldrich) to the organic solvent used for spray-drying. Micronized surfactant particles were produced using GEA Niro PSD-1 or (Niro Inc., Copenhagen, Denmark) or Buchi B-290 mini (Buchi Corporation, New Castle, DE, USA) spray-dryers. After spray-drying, the DP B-YL: Trehalose surfactant was loaded into size 00 capsules (30 mg per capsule) and packaged in heat-sealable pouches with desiccant. The surface activity of the B-YL: Trehalose surfactant was quantified using captive bubble surfactometry. All reagents were stored at 7 °C and 40% humidity and kept in blister packaging before testing.

### 2.2. Low-Flow Aerosol Chamber (LFAC)

The B-YL: Trehalose surfactant was aerosolized with a low-flow aerosolization chamber (LFAC) designed by Acorda Therapeutics Inc., based on simplicity of design and use, minimum number of parts, and low cost of goods and manufacturing. The LFAC is a cylindrical chamber/inhaler with several holes at one end that accommodates a perforated DP capsule and dispenses aerosol into the inhalation pathway via the nasal prong interface (Figure 1). The LFAC does not require auxiliary electricity or compressed medical gases to operate. It uses a 60 mL bellows on the posterior end of the LFAC to generate the airflow flow (~8 L/min) necessary to spin the drug capsule. The frictional and centrifugal forces of the spinning capsule and turbulent flow passing through the punctured capsule cause shearing (milling) and powder dispersion from the LFAC into primary particles. A series of one-way valves and a three-way stopcock at the inlet of the inhaler allow the bellows to reinflate from the atmosphere without entraining aerosol back into the LFAC capsule or inhaler chamber (Figure 1). Since aerosol generation is dependent on intermittent actuation of the bellows, aerosol output from LFAC can be manually timed to coincide with patient breathing efforts for synchronized inspiratory delivery of surfactant. The LFAC outlet consists of a 2.0 mm ID hose barb/Luer fitting resistor that can be attached to a nasal interface via injector outlet tubing.

### 2.3. Aerosol Particle Sizing

A multi-staged next-generation impactor (NGI, Copley Scientific, Colwick, UK) was used to characterize aerodynamic particle size distribution (APSD) and classify the DP surfactant into respirable size fractions. The impactor uses seven individually staged gravimetric particle trays that are recessed to accommodate a 45 mm glass fiber (GF) filter substrate (Millipore Sigma, Burlington, MA, USA) for aerosol collection. GF filters were pre-conditioned with Molykote^®^ silicone spray, dried, and weighed with an AX205 Delta Range Lab balance (Mettler Toledo, Columbus, OH, USA). A vacuum and frit resistor (S/N 511197-9, Cole Palmer, Vernon Hills, IL, USA) was used to maintain nominal impactor flow (15 L/min) and confirmed with a TSI 5200 Flow and Pressure Analyzer (TSI Industries, Shoreview, MN, USA). An NGI leak test was performed per the manufacturer’s specifications before testing. The LFAC was pre-loaded with a punctured B-YL: Trehalose capsule (30 mg), and the bellows was actuated 50 times. Aerosol was dispersed from the LFAC outlet adaptor and injector tubing into the NGI inlet. Following nebulization runs (*n* = 6), the aerosol drug mass (µg) deposited on the staged filters was weighed with a balance. MMAD and geometric standard deviation (GSD) were calculated based on gravimetric changes in filter mass (µg) delivered to the different NGI stages following nebulization. The fine particle fraction (FPF) was the proportional aerosol mass delivered to the NGI with an MMAD <5.4 µm. Particle size analysis was performed with Inhalytics software version 1.0 (Copley Scientific, Colwick, UK), which is 21 CFR Part 11 compliant and meets the requirements of United States Pharmacopoeia (USP) 43 and European Pharmacopoeia (Ph Eur) 10.0.

### 2.4. Preterm Infant Nasal Airway and Lung Model 

A neonatal nasal airway and lung model was configured to generate the realistic, spontaneous breathing parameters of a 1200 g preterm infant affixed with a realistic nasal airway cast, bCPAP, surfactant delivery system, and different nasal CPAP interfaces (see Figure 1, Figure 2 and Figure 3). A high-fidelity neonatal lung model (ASL 5000 breathing simulator, Ingmar Medical, Pittsburg, PA, USA) was used to evaluate nebulizer operational compatibility when the LFAC system was integrated with bCPAP for surfactant administration. This model acquires real-time digital pressure and volume measurements (500 Hz) from within the breathing cylinder, and the internal component cannot be exposed to aerosol. Nebulizer efficiency and delivered dose studies were conducted using a Harvard Small Animal Rodent Ventilator (model 683, Harvard Apparatus, Holliston, MA, USA) to simulate breathing. This mechanical model has an externalized piston and cylinder assembly that can be removed for cleaning. Both lung models were configured with baseline preterm infants’ lung mechanics and spontaneous breathing parameters, with compliance of 1.0 mL/cm H_2_O, airway resistance of 100 cm H_2_O/L/s, respiratory rate of 50/min, and V_T_ of 8 mL (~7 mL/kg) [18,19]. A 3D-printed anatomic nasal airway model was constructed using data acquired from the computed axial tomographic scan of a preterm infant of ~30 weeks gestational age and is described elsewhere [20]. The CT scan included the area from the nostrils to the nasopharynx and was connected to the lung model using a low dead space tracheal adaptor which together approximated the internal volume and resistance of the preterm nasotracheal regions which we refer to simply as “nasal airways” in this manuscript. An infant chest-rise analog, consisting of a small silastic bladder and simulated infant chest wall, was intermittently inflated with an air compressor and timer that was turned on and off based on the flow signal (voltage) from the lung model. A single operator (RMD) was able to visualize chest rise and fall and manually depress the LFAC bellows to appropriately time (i.e., synchronize) LFAC actuation and aerosol delivery with lung model inhalation.

### 2.5. Bubble CPAP System and Nasal Airway Interfaces

Bubble CPAP is widely used in LMICs, and pressure is created with the underwater seal [21]. The bCPAP system consisted of a water column (pressure generator), inspiratory and expiratory patient circuits, and an MR850 heated humidifier (Fisher & Paykel, Healthcare Inc. Irvine, CA, USA). The CPAP level was set to 6 cm H_2_O, and system flow was adjusted (6–8 L/min) to maintain constant bubbling throughout the respiratory cycle. Before testing, bCPAP system pressures and flows were confirmed with a calibrated analyzer (TSI Inc., Shoreview, MN, USA). 

We selected nasal prong interfaces commonly used to deliver CPAP (*n* = 3), and cannula prototypes (*n* = 2) specifically designed for DP aerosol delivery. The RAM Cannula (Neotech Products, LLC, Valencia, CA, USA) and Hudson Prongs (Hudson-RCI, Temecula, CA, USA) were adapted to streamline aerosol delivery with bCPAP. The RAM cannula (Figure 2A) and bCPAP patient circuit were connected in series using an elbow adaptor with a perpendicular aerosol injection port placed within 2 cm of the cannula tubing entry point. The Hudson RCI cannula (Figure 2B) was adapted to emit aerosol from the LFAC injector into the CPAP flow via a Luer-lock fitting on the inspiratory manifold to monitor airway pressure. The GINEVRI prong interface (Rome, Italy) was attached in series with a specialized aerosol connector (AFECTAIR^®^, Windtree Therapeutics Inc., Warrington, PA, USA) with an internal channel designed to separate the fluidic paths of aerosol from CPAP flow at the nasal interface [22]. The LFAC outlet tubing was inserted through the center channel within 2 cm of the prong inlet (see Figure 2C). The Neotech Aerosol Delivery Prong (Prototypes 1 and 2, Neotech Valencia, CA, USA) was designed to incorporate a streamlined aerosol flow channel mostly separated from the CPAP flow (Figure 3A–C). Prototype 2 includes angled tubing at the expiratory outlet (see Figure 3B) to direct more aerosol to the nasal airway opening. Descriptive computational fluid dynamics (CFD) analysis was performed with 3D models of the Neotech Aerosol Delivery Prong using Solidworks (2021 Flow Simulation module) to evaluate in silico particle behavior and interaction with CPAP flow using the prototype prongs. The analysis was based on density, dynamic viscosity, surfactant-specific heat, thermal conductivity density, MMAD of DP B-YL: Trehalose surfactant combined with lung model flows, CPAP flow, and pressure, and LFAC flow output and is described in greater detail in Figure 3C. All nasal interfaces were inserted 2 cm into the openings of the nasal airway model and affixed with a hydrocolloid securement barrier (Cannulaide, Sun Med, Grand Rapids, MI, USA) to provide an occlusive fit and prevent the leakage of aerosol and CPAP. 

### 2.6. LFAC Operational Compatibility with bCPAP 

Experiments were performed to evaluate operational feasibility and compatibility between the LFAC delivery and bCPAP. In consideration of the LFAC flow output (~8 L/min) generated by the bellows for inspiratory drug delivery, we sought to determine whether additive flow combined with bCPAP and superimposed on spontaneous breathing potentially increases the risk for pulmonary overdistention or volutrauma. Acceptability criteria were established a priori to identify pressure and volume limits commonly associated with increased risk for pulmonary injury and inflammation in preterm infants. Testing conditions that resulted in a peak inspiratory pressure (PIP) >25 cm H_2_O, absolute V_T_ >12 mL (>10 mL/kg) [23,24,25], and mean airway pressure ≥2 cm H_2_O above CPAP [26] during aerosol therapy would preclude the use of selected prong interfaces in subsequent delivered dose aerosol studies. Baseline measurements of delivered PIP, inspiratory V_T_, and mean airway pressure (i.e., CPAP) were measured within the ASL 500 lung model during simulated bCPAP of 6 cm H_2_O and the various interfaces (Figure 2 and Figure 3) without LFAC activation. A punctured sham capsule (air-filled, no powder) was placed into the LFAC chamber. Measurements were repeated with manual LFAC actuation (*n* = 50) with manual bellows deflation timed to coincide with the lung model chest-rise analog inflation. Descriptive data were obtained from within the ASL test lung and reported as mean ± SEM.

### 2.7. Nebulizer Efficiency and Delivered Dose Studies

The experimental setup shown in Figure 1 was used for aerosol studies, but the ASL 5000 was interchanged with the Harvard Apparatus lung model to facilitate cylinder cleaning following exposure to aerosol and humidity. We evaluated LFAC nebulizer efficiency based on estimates in (1) LFAC emitted dose; (2) residual drug mass within the capsule following nebulization; (3) proportion of emitted dose and capsule loading dose delivered to the nasal airway and lung model; and (4) estimated depositional loss within the LFAC, prongs interfaces, and bCPAP system between the different CPAP interfaces. The laboratory testing conditions were a relative humidity (RH) of 40% and a temperature of 20 °C. The lung model was attached to the bCPAP system via the nasal airway model using the different nasal interfaces and a nasal barrier. Initially, the bCPAP MR 850 humidifier reservoir was filled with sterile water with the heater in the “OFF” position. This “passive” (non-heated) humidity condition produced a RH of ~30%, confirmed using a hygrometer (Thermo Fisher Scientific, Waltham, MA, USA) placed in the inspiratory limb. The lung model filter and nasal airway cast were sprayed with silicone Molykote^®^, dried, and weighed prior to nebulization. The lung model filter was placed within a low dead space filter housing and attached to the nasal airway. A series of one-way valves was placed between the filter housing and lung model to prevent inhaled aerosol on the GF filter from recirculating back into the bCPAP system (Figure 1). 

A 30 mg capsule of DP B-YL: Trehalose surfactant was punctured and placed into the LFAC chamber. Aerosol was generated with bellows deflation timed to coincide with the inflation of the lung model chest-rise analog (*n* = 50 actuations/run). Each of the CPAP interfaces (*n* = 5) was tested over six runs (*n* = 6) for a total of 30 measurements for the passive humidification condition. The difference in the DP capsules’ gravimetric mass (µg) between pre- and post-nebulization and following the cleaning of the capsule was used to determine the nominal, residual, and emitted DP surfactant dose from the LFAC nebulizer. Following aerosol delivery, the nasal airway and lung model filters were reweighed and the changes in mass delivered at each location represented the depositional loss to the upper airway and estimated inhaled lung dose, respectively. Each was referenced to the total emitted dose of the LFAC for each run. Depositional losses to the LFAC, interfaces, and bCPAP system could not be measured directly and were estimated based on the mass balance differences between the capsule loading and residual doses and cumulative mass deposited in the nasal and lung model. All devices were cleaned with sterile water and dried in a vacuum following nebulization.

The deposited mass within the nasal airway and GF filter represented the entirety of the solid mass of the active B-YL: Trehalose surfactant used in our gravimetric assay. Prior to testing, the gravimetric assay was validated by comparing the mass recorded by the balance in 30 mg capsules (49 weight%) with the phosphatidylcholine (PC) content using a biochemical PC assay (ELISAs, ab83377, Abcam, Cambridge, MA, USA). The colorimetric mass values were measured at OD450 nm with wavelength correction set to 540 nm with a multi-mode microplate reader (SpectraMax M3, Molecular Devices, San Jose, CA, USA). There was a high correlation between gravimetric and biochemical PC assay methods (*r* = 0.94).

The final experiments were designed to evaluate aerosol delivery efficiency with active heated and humidified bCPAP gases with select nasal interfaces. The MR 850 humidifier reservoir was filled with sterile water with the heater in the “ON” position to obtain “active” heated humidification with a relative humidity of 100% and temperature of 39 °C (confirmed with the hygrometer). The lung model chamber was heated to 37 °C with an internal heat controller to heat exhaled gas through the nasal airway and prevent condensation from forming within the lung cylinder. The internal components of the bCPAP system, lung model, and interfaces were exposed to “active” heated humidity for 10 min prior to nebulization. Nebulization was performed with Neotech prototype 2 and Ginevri (Afectair) prong interfaces (*n* = 5) using the same methods described above. Gravimetric analysis resulted in instability due to the high moisture content. Following nebulization, the nasal airway and lung filters were eluted with 10 mL of 0.1 M NaCl and 0.1 M NaHCO_3_. The recovered mass was quantified using PC assay and referenced to the emitted and total PC mass (49 weight%) of the capsules. 

### 2.8. Statistical Analysis

The primary outcome of the study was to evaluate nasal airway deposition and lung dosage of DP surfactant aerosol during bCPAP among five different nasal interfaces. Reported values were summarized as means ± standard deviation (SD) in text and figures. Comparison of multiple groups during delivered dose studies was evaluated by one-way ANOVA with Tukey’s post-hoc test. Data for humidity testing were carried out by comparing passive humidity to active humidity conditions and analyzed with a two-tailed unpaired *t*-test. Criterion for significance was *p* < 0.05 for all comparisons. We used GraphPad Prism version 9 (GraphPad Software, Boston, MA, USA) for statistical analysis. Measurements of particle size and the operational compatibility of DP surfactant delivery were analyzed using description statistics. The mean pressure and volume delivery outcomes were compared against the acceptability limits established a priori.

## 3. Results

### 3.1. Aerosol Particle Size Distribution

The DP surfactant aerosol particles generated at the LFAC outlet had a MMAD of 3.68 µm and GSD of 1.78. The FPF (<5.4 µm) was 71% and 41% based on the cumulative mass deposited in the NGI and the capsule loading dose, respectively. 

### 3.2. LFAC Operational Compatibility with bCPAP

The effects of peak pressure and volume delivery in the lung model are shown at baseline (no bellows) and with bellows actuation with different nasal prong interfaces and bCPAP in Figure 4. There were several outlier breaths with the Neotech (Prototype 1) interface that exceeded the upper volume limit of 12 mL (10 mL/kg), but the mean value remained below the acceptable V_T_ limit. The Neotech (Prototype 2) and all other interfaces delivered V_T_ and pressure measurements that were well within the established clinical limits when the bellows flow was superimposed on spontaneous breathing with bCPAP.

### 3.3. Nebulizer Efficiency and Delivered Dose 

The LFAC nebulizer output was highly efficient with a mean emitted DP surfactant aerosol dose ≥ 90% and low residual ≤10% medication remaining in the capsule for all interfaces tested (Figure 5). DP aerosol surfactant deposition in the nasal airway and lung model filter were highly dependent on the type of nasal interface and bCPAP gas humidity levels (*p* < 0.05) (Figure 6). 

In the passive humidity condition, the Ginevri (Afectair) interface resulted in greater nasal deposition than any of the other interfaces (*p* < 0.001) and a higher delivered lung dose than the RAM or Hudson interfaces (*p* < 0.001) (Figure 5 and Figure 7). The RAM and Hudson interfaces had greater nasal airway than lung filter deposition and less aerosol delivery to the nasal airway and lung than the other nasal interfaces. The Neotech interfaces showed lower nasal deposition than the Ginevri interface with 30% and 40% lung filter delivery with Neotech prototypes (Version 1) and (Version 2), respectively (Figure 6 and Figure 7).

During active humidification of bCPAP gases (99% RH), the emitted dose and residual DP drug mass was not impacted (<10%) (Figure 7). However, active humidity applied with Ginevri (Afectair) and Neotech (Prototype 2) interfaces resulted in less drug deposition onto the lung filter than with passive humidity (*p* < 0.001) (Figure 6 and Figure 7). The depositional losses occurring within the interfaces, bCPAP, connectors and hoses, and medication delivery system (LFAC) with this condition were >80% of the total DP surfactant dose in most cases (Figure 7). The reduced deposition in the lung filter resulted from the agglomeration of the DP surfactant within the LFAC outlet, nasal prongs, and nasal airway. In some cases, the nasal airway openings and flow channels became partially or totally occluded with a pasty glue-like material from the combined DP aerosol and humidity (see Figure 6A–C). Only a minimal amount of medication could be eluted and recovered from the nasal airway for analysis, resulting in lower nasal deposition values than were observed with passive humidification.

## 4. Discussion

Our findings indicate that a high dose of DP surfactant particles can be delivered beyond the upper airway to the lung without increasing the potential risk for lung injury during concurrent aerosol and bCPAP treatment in a preterm infant model. However, delivery efficiency was highly dependent on the type of nasal prongs and the gas humidity levels provided with bCPAP. Specialized prongs designed to disperse aerosol proximal to the nasal airways resulted in the lowest nasal airway deposition and greatest inspired drug dose (30–40%) of surfactant. The use of a heated humidified bCPAP gas source resulted in notable DP surfactant adhesion and agglomeration within the prongs and nasal airways and compromised drug delivery and safety, raising significant safety concerns for preterm infants.

### 4.1. Safety and Efficacy

A recent publication by the World Health Organization stated that “aerosolized surfactant and its delivery system should be as safe and efficacious as conventional surfactant formulations” for use in preterm infants in LMICs [27]. In vitro lung models are important for testing patient safety and should be applied prior to introducing novel aerosol drug delivery systems with neonatal positive pressure devices in newborn infants. We showed in a preterm infant nasal airway/lung model that the intermittent LFAC inhaler actuations and flow output superimposed on spontaneous inspiratory efforts with bCPAP resulted in negligible increases lung volume and pressures and did not exceed clinically important thresholds for lung protection. The additive air flow from the LFAC did not increase the delivered CPAP levels by more than 1 cm H_2_O (Figure 3). This represents an important development in pulmonary aerosol drug delivery for preterm infants because it preserves the beneficial effects of bCPAP. The pressure generated by the bellows at the LFAC inlet (25 cm H_2_O) was attenuated by the LFAC chamber and outlet (10 cm H_2_O). The inflation pressure was further attenuated by the CPAP interfaces, breathing circuits, and release from the water column during bCPAP. The bCPAP water-seal is considered a “pure threshold resistor” [28]. The LFAC flows that generated system pressures greater than the set point CPAP level (6 cm H_2_O) were diverted to the CPAP hoses and water column and released to the atmosphere. The natural pressure release or “pop-off” with bCPAP is less likely to occur with LFAC when applied with other forms of CPAP that employ resistive valves to maintain CPAP.

The LFAC inhaler system applies aerosol to the nasal prongs by manually deflating a 60 mL bellows bottle. This simplistic method for generating aerosol for inhalation extends on the practice of applying positive pressure with a manual resuscitator (bag) to preterm infants, which is commonly performed in newborns with weak or ineffective inspiratory efforts in LMICs. In contrast to manual ventilation, the infant remains on bCPAP and nasal aerosol delivery can be provided through prongs without increasing the positive pressure to the infant. Of course, aerosol delivery efficiency relies on coordinated efforts to appropriately synchronize aerosol dispersion with the infant’s inspiration. This may be difficult as the respiratory rate of newborn infants is about 40/min and increases during respiratory distress.

### 4.2. Particle Size

We found that DP surfactant aerosols generated with a LFAC have a MMAD of ~3.8 µm. A small particle size is desirable to prevent nasal impaction and optimize pulmonary delivery in preterm infants. However, there is little data about the optimal particle size for surfactant aerosols in preterm infants that enhances deposition in the acinar lung regions without being exhaled. Using a 30-week gestational age preterm infant model, Clark [29] reported that lung deposition (% of inhaled) of inhaled aerosol particles with a MMAD/GSD of 2.5 µm/2.5, 3.0 µm/1.75, and 4.0 µm/1.0 amounted to, respectively, 23.6%, 29.3%, and 36.8%, i.e., lung deposition increases with increasing MMAD and decreasing GSD. These data are comparable to our findings when DP surfactant was given using Neotech prongs during bCPAP with passive humidity. Clark concluded that medical aerosols can be efficiently delivered to newborn infants via nasal CPAP if the MMAD is ~2.5–3.0 µm, the GSD is 1.5–2.0, and breath synchronization is combined with aerosol delivery in the first 80% of inspiration. Future studies are needed to determine optimal particle sizes when applying inhaled drugs to preterm infants receiving bCPAP with and without heated-humidified gases.

### 4.3. Lung Dose

Lung deposition of aerosols is challenged by their particle size, nebulizer type, and nebulizer placement location within the nasal CPAP system. Nebulizers that generate aerosol throughout inspiration and expiration result in high expiratory losses to the upper nasopharyngeal airways and CPAP system. We found that small, inhaled synthetic DP surfactant particles can be generated with a simple low-cost inhaler chamber and bellows bottle and applied to nasal prongs and bCPAP without increasing the risk for lung injury. This resulted in low nasal airway deposition and high inhaled lung dose (30–40%) with passive humidification, but only when therapy was applied with prongs that were specifically designed to separate the bCPAP flow from the aerosol flow. 

We observed small increases in airway pressure and lung volume during DP delivery with LFAC that may have resulted in minimizing aerosol losses by enhancing particle movement into the distal airways during inhalation. Amirav et al. have previously shown that small increases in tidal volume result in large increases in the lung delivery of aerosols in infants’ nasal airway models [30]. The high efficiency of DP delivery with LFAC may therefore be related to these small increases in tidal volume and airway pressure delivery (>2 cm H_2_O) that overcome dead space and facilitate aerosol movement and penetration into the distal airways. 

### 4.4. Nasal Prongs

We found large differences in aerosol efficiency between the different CPAP prong interfaces configured for drug delivery. Using off-the-shelf interfaces (Hudson and RAM) with a “mainstream” aerosol administration delivered a negligible inhaled mass (<2%) to the lung model. The lack of aerosol deposition in the nasal and lung filters and the low residual drug remaining in the capsule (~10%) with RAM and Hudson prongs indicate that the entirety of the emitted aerosol was lost in the nasal prongs and bCPAP system flow. Interestingly, we observed fine aerosol particles exiting the bCPAP water-seal and exhaust port or depositing in the water column, creating foam. 

DP aerosol delivery was most efficient when using interfaces that emit aerosol proximal to the nasal openings with specialized aerosol flow channels that mitigate cannula losses and dilution of particles with the bCPAP source gas. The Neotech prototype prongs, designed with specialized channels that effectively decouple aerosol flow from bCPAP flow and provide a more direct route for nasal delivery, enhanced the delivered lung dose to 30–40% with only modest nasal airway losses (<10%). Based on comparisons with previous bench studies of nebulized surfactants, this is the highest efficiency published to date for inhaled dose. Interfaces that circumvent the effects of CPAP system flow with a “sidestream” aerosol delivery method may reduce the residence time of the aerosol from being exposed to humidity, limit the hygroscopic growth of aerosol particles, and prevent the entrainment of aerosol into the CPAP flow and prevent expiratory drug loss.

### 4.5. Nebulizer

Experimental nebulizers designed for liquid aerosol of surfactant delivery typically generate a constant output of the drug during the respiratory cycle that may be less efficient for pulmonary drug delivery than breath-synchronized nebulizers designed to time aerosol delivery with the patient’s inspiratory phase [31,32,33]. Moreover, constant output aerosol nebulizers have been shown to result in high drug loss in the upper airways (~80%) [31], poor lung delivery, and 99% of aerosolized surfactant depositing in the expiratory tubing [8]. The high expiratory losses contribute to increased nasal resistance and congestion, higher work of breathing, preventing surfactant passage to the acinar lung units, and poor equipment performance or malfunction, but also require multiples of the standard instilled surfactant dose to have a therapeutic effect. Jorch et al. [34], using a continuous jet nebulizer to deliver liquid surfactant, were among the few investigators reporting a promising clinical response, but concluded that aerosol doses up to four-fold larger than liquid intratracheal instillation made aerosol delivery impractical in terms of both cost and time of administration in neonatal RDS. These limitations in aerosol therapy make it difficult to implement constant output nebulizers for surfactant administration in LMICs, especially when considering the economic and safety implications. Better in vitro and in vivo results have been obtained in the current study or when using vibrating mesh with breath-synchronized nebulizers and are the way forward in improved lung dosing of surfactant. A combination with Neotech prototype aerosol delivery prongs may help to reduce nasal deposition and further improve pulmonary deposition. However, prevention of atelectasis by appropriate use of bCPAP must be the first step before delivering aerosolized surfactant in this complicated treatment model.

### 4.6. Humidification

DP aerosol surfactant delivery with an active heated humidified bCPAP gas source resulted in the notable adhesion and agglomeration of surfactant in the nasal prongs and airways, reduced lung delivery efficiency, and significant clinical safety concerns for preterm infants at risk for airflow obstruction during bCPAP. There are concerns about hygroscopicity, solid state stability, and aerosol efficiency when applying spray-dried inhalation powders into a heated/humidified bCPAP system. The exposure of emitted DP aerosol particles to condensed water vapor within internal plastic surfaces (e.g., interfaces and delivery tubing) can result in agglomeration and high depositional loss of aerosol in systems with high RH from active humidifiers. Thus, the hygroscopic growth of aerosol particles in ventilator circuits can reduce medication delivery by as much as 40%. In some cases, this resulted in partial or total nasal airway occlusion of with pasty glue-like material in the 3D printed model (Figure 6). There is a generalizable gap in knowledge on whether a heated humidified gas source is required for neonatal noninvasive respiratory support. As is well known in intubated patients where the upper airways are bypassed with an endotracheal tube, a heated and humidified gas source is required when applying mechanical ventilation. Heated humidity with an active humidifier is commonly applied in neonates on nasal CPAP in resourced settings, but this often results in excessive accumulation and aspiration of condensed water vapor in the hoses and prongs [35]. In sub-Saharan Africa approximately 4–10% of neonatal units that apply noninvasive support can provide humidified blended oxygen with medical air to all infants [36]. This raises the question of whether passive humidification of bCPAP gases should be preferred over active heating and humidification when delivering aerosolized surfactant, especially considering that most babies supported in delivery-room settings are supported using dry (anhydrous) oxygen with manual resuscitation and CPAP. Short-term interruption of heated humidity to circumvent its negative effects on aerosol delivery of surfactant has not been well described in nasally breathing infants receiving dry medical gases on noninvasive support and cannot be suggested at this time. 

### 4.7. Limitations

We used a single preterm infant nasal airway and lung model as we were limited in our ability to acquire models that would span the individual neonatal sizes from the child cohort that is being treated with surfactant for RDS in an intensive care setting. A single operator manually activated the bellows based on the visual chest rise and fall using an infant chest-rise analog. The delivery efficiency of DP surfactant aerosols may be heavily influenced by the operator’s ability to observe chest rise and appropriately time LFAC actuation with inhalation, especially in tachypneic newborns with severe RDS. Efforts should be made in the future to fully automate and synchronize inspiratory delivery of DP surfactant aerosols so that infants can receive more effective dosing. Our results with active humidity should be approached with trepidation because they are limited to the use of non-heated anatomic airway models which are more prone to condensation and “rain-out” and powder agglomeration than newborn human infant airways. Evaluation of the hygroscopic growth and sedimentation of DP aerosol particles should be explored in greater detail before this therapy can be implemented in the clinical setting. The filter technique employed to measure inhaled lung dose only estimates the total surfactant dose delivered to the lungs and allows no conclusion to be reached about the amount that reaches the lower airways where it is active.

Future testing is required to address particle size related to hygroscopic particle growth when using a heated/humidified gas source or when capsules are exposed to different environmental factors. Next steps should also include testing in the lung model of higher fill doses (50 instead of 30 mg) of B-YL: Trehalose surfactant to determine efficiency and nasal deposition. A pre-clinical study in surfactant-deficient rabbits would be useful for evaluating dosing responses with 50 mg capsules and the potential risk using selected aerosol flow-generating procedures, nasal interfaces, and bCPAP systems prior to being used in humans.

## Figures and Tables

**Figure 1 pharmaceutics-15-02368-f001:**
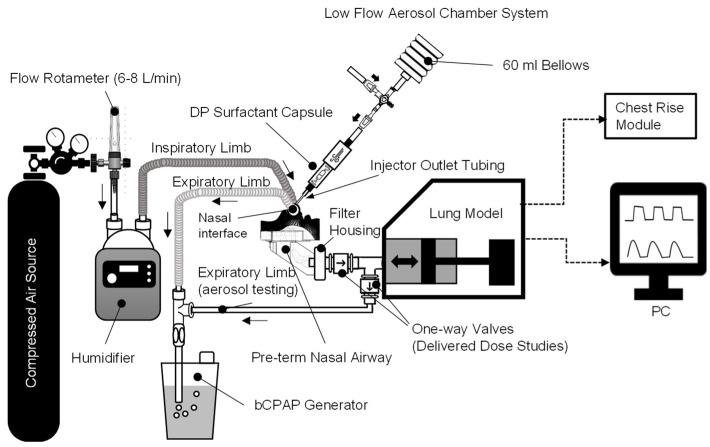
Experimental Set-up and Low-Flow Aerosol Chamber (LFAC) System. The B-YL: Trehalose surfactant was aerosolized with a LFAC, a cylindrical chamber/inhaler with several holes at one end that accommodates a perforated DP capsule and dispenses aerosol into the inhalation pathway via the nasal prong interface. The LFAC does not require auxiliary electricity or compressed medical gases to operate. It uses a 60 mL bellows on the posterior end of the LFAC to generate the airflow flow (~8 L/min) necessary to spin the drug capsule and release aerosol. A series of one-way valves and a three-way stopcock allow the bellows to reinflate without entraining aerosol back into the LFAC capsule or inhaler chamber. The black arrows show the direction of aerosol flow through the LFAC during bellows deflation (inhalation) and entrainment of ambient flow which allows the bellows to reinflate (exhalation) with the use of one-way valves.

**Figure 2 pharmaceutics-15-02368-f002:**
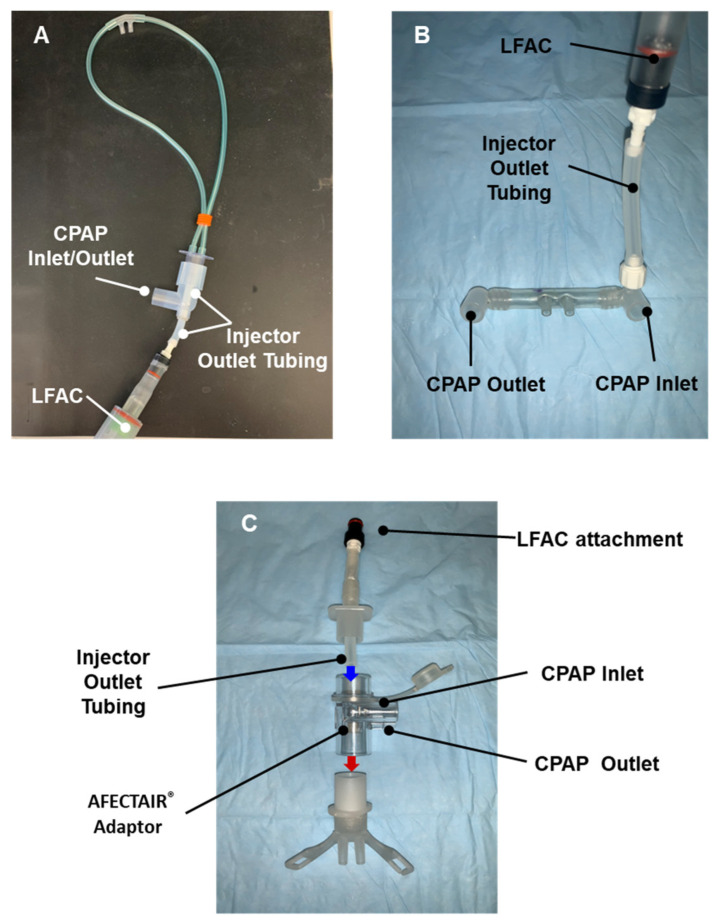
Bi-nasal short prongs configured to provide CPAP and intermittent aerosol delivery. The RAM Cannula (Neotech Valencia, CA, USA) and Hudson Nasal Prongs (Hudson-RCI, Temecula, CA, USA), two commonly used interfaces, were adapted to provide specialized flow channels for medication delivery to disperse the aerosol plume directly into the bCPAP source gas flow. The RAM cannula (**A**) 15 mm adaptor was attached to an elbow adaptor with a perpendicular CPAP port and small port which allowed Injector Outlet Tubing to be inserted within 2 cm of the cannula tubing openings. The Hudson cannula (**B**) integrates a port for measuring pressure which was used in this testing to disperse aerosol into the CPAP flow by attaching the LFAC injector between the CPAP inlet and nasal prongs using a Luer fitting. The Ginevri prongs (Ginevri srl, Rome, Italy) were adapted to the AFECTAIR^®^ connector (**C**) that has an internal channel designed to separate the fluidic paths of the aerosol and CPAP flow. The LFAC aerosol inlet was inserted into the internal channel of the AFECTAIR connector via a 15 mm adaptor (see blue arrow) and the patient interface port was attached to the Ginevri prongs inlet adaptor (see red arrow).

**Figure 3 pharmaceutics-15-02368-f003:**
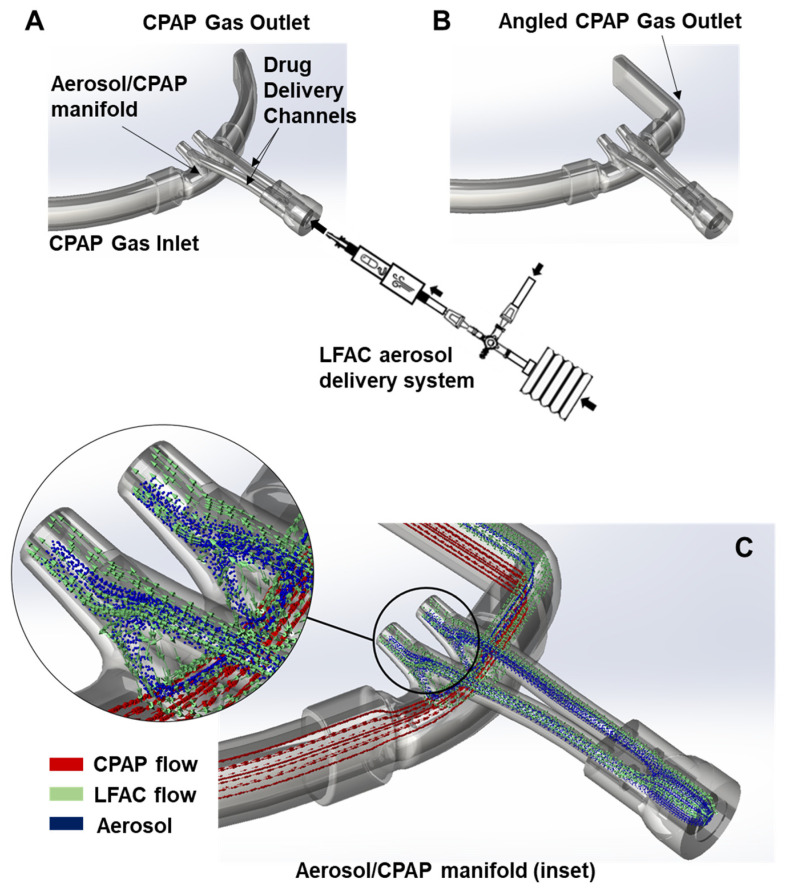
Neotech Aerosol Delivery Prongs. The Neotech prototype cannulae are designed with a perpendicular access channel to improve aerosol delivery efficiency and safety. The aerosol/CPAP patient manifold decouples LFAC and CPAP flow pathways so that the mixing of aerosol with CPAP flow occurs within a short timeframe to minimize aerosol dilution and losses to the expiratory limb by bCPAP flow. The major physical differences between Neotech prototypes 1 (**A**) and 2 (**B**) is the addition of an angled expiratory manifold outlet with prototype 2. We speculated that the small increase in downstream resistance with an angled outlet could enhance aerosol streaming into the nasal prongs and provide a higher inhaled surfactant dose. Computational fluid dynamics (CFD) helped to illustrate the potential behavior of incoming LFAC aerosol flow through the drug delivery channels and the boundary condition that prevents the bCPAP flow from mixing with aerosol in the manifold during inhalation (**C**). The small internal diameter of the parallel drug delivery channels attenuates pressure generated by LFAC (~25 cm H_2_O) resulting in high gas velocity at the patient manifold. Upon inhalation, airway pressure at the prong outlet (and lung) decreases in relation to the bCPAP level (6 cm H_2_O), and when the LFAC is timed with inhalation, aerosol enters the nasal airway and the flow and pressure exceeding the set point CPAP level (~6 cm H_2_O) is diverted to the the bCPAP circuit and released within the water column, preventing over pressurization in the lungs. If the is LFAC is mistimed with the patient effort, any increase in PIP > CPAP would be minimal because LFAC flow is released through the water-seal, preventing excessive pressure and volume delivery to the lungs during medication delivery. At end-inhalation, LFAC flow ceases and back-pressure within the drug delivery channels increases, and the patient can exhale through the prongs, patient manifold, and bCPAP column. The black arrows show the direction of aerosol flow through the LFAC during bellows deflation (inhalation) and entrainment of ambient flow which allows the bellows to reinflate (exhalation) with the use of one-way valves.

**Figure 4 pharmaceutics-15-02368-f004:**
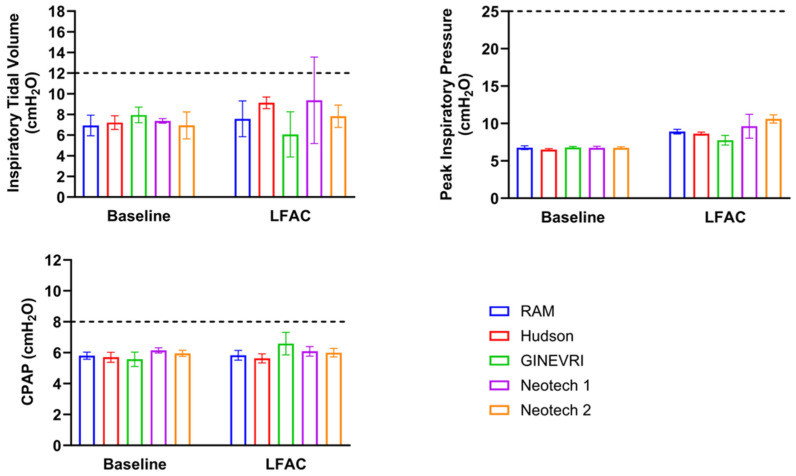
LFAC compatibility during aerosol surfactant delivery with bCPAP and different airway nasal interfaces. All interfaces had an acceptable level of safety on delivered tidal volume, peak inspiratory pressure, and PEEP within the test lung. LFAC: low-flow aerosolization chamber; PEEP: Positive end expiratory pressure. *n* = 20 breaths/group. Values are means ± SD. The dotted lines represents the maximum acceptable pressure and volume limits commonly associated with increased risk for pulmonary injury and inflammation in preterm infants.

**Figure 5 pharmaceutics-15-02368-f005:**
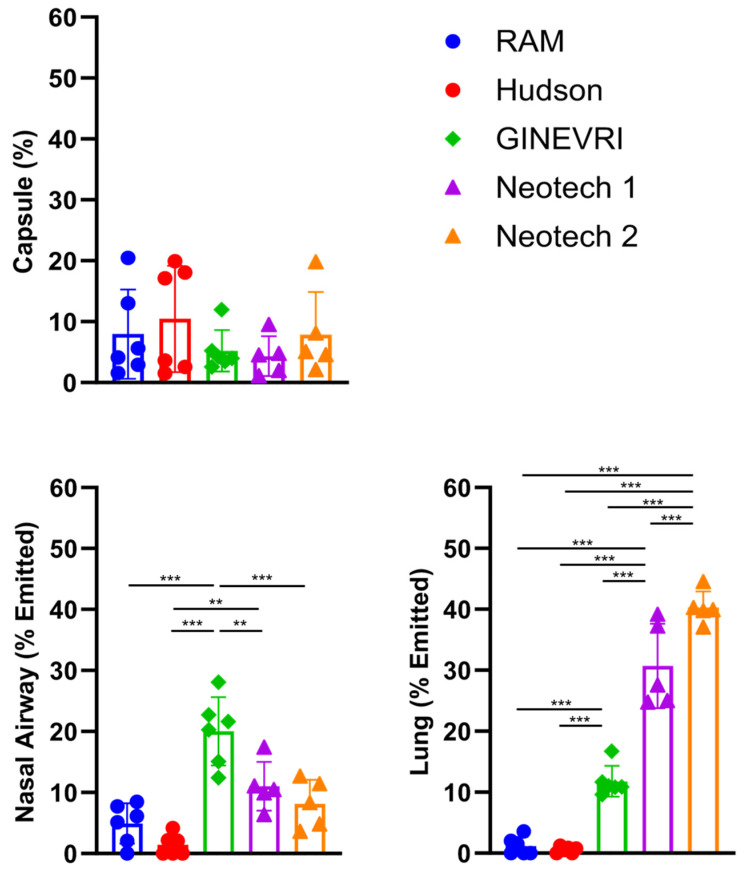
Surfactant aerosol deposition in the nasal airway and lung model with LFAC and bCPAP with passive humidification. Main-stream interfaces (RAM and Hudson) showed low aerosol delivery deposition to nasal airway and lung, while side-stream interfaces (Ginevri and 2 types of Neotech) showed greater aerosol delivery deposition to nasal airway and lung, especially for lung delivery with both Neotech prongs. *n* = 6/group. Values are means ± SD. **, *p* < 0.01; ***, *p* < 0.001. Values were compared using one-way ANOVA with post-hoc Tukey multiple comparison. Capsule: residual powder in capsule after procedure. Nasal Airway: aerosol deposition in the nasal passages, pharynx, larynx, and tracheal adaptor. Lung: aerosol deposition captured within the filter at the distal trachea region.

**Figure 6 pharmaceutics-15-02368-f006:**
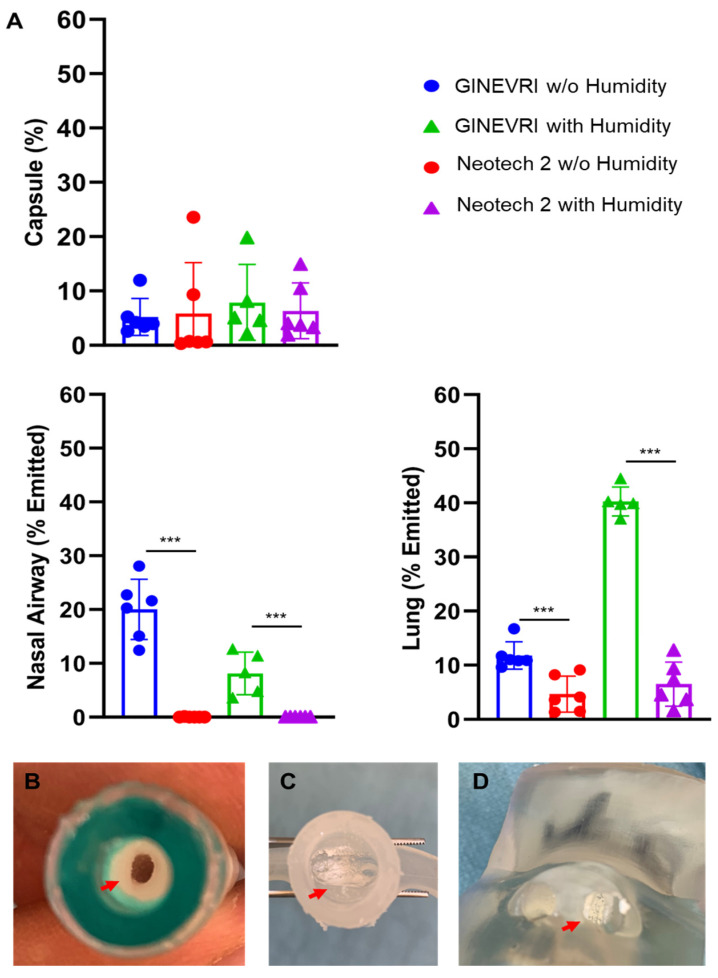
Surfactant aerosol deposition in the nasal airway and lung model with LFAC and bCPAP with active heating and humidification. (**A**) Humidity-resulted in low deposition to nasal airway and lung. *n* = 5–6/group. Values are means ± SD. ***, *p* < 0.001. Values were compared using the two-tailed unpaired *t*-test between without (w/o) and with humidity in each interface. There was high deposition and agglomeration of DP surfactant within the LFAC delivery tubing (**B**), nasal prongs (**C**) and nasal airway opening (**D**). Red arrow: powder deposition.

**Figure 7 pharmaceutics-15-02368-f007:**
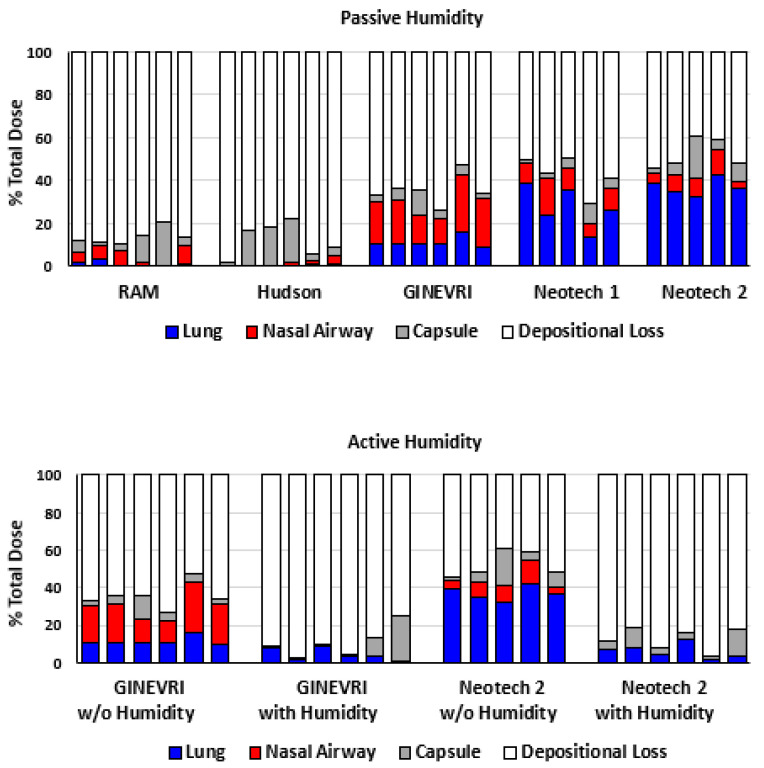
Mass balanced showing proportion of drug mass at each location referenced to the capsule dose for passive and active humidification of bCPAP gases. Lung: aerosol deposition captured with filter at trachea region. Nasal Airway: aerosol deposition in the nasal passages, pharynx, larynx, and tracheal adaptor. Capsule: residual powder in capsule after procedure. Depositional loss: aerosol depositing within the LFAC, nasal prong interfaces, bCPAP system.

## Data Availability

Data will be available upon request.

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
