# Peer review of "Evaluation of a Novel Dry Powder Surfactant Aerosol Delivery System for Use in Premature Infants Supported with Bubble CPAP"

_pharmaceutics, 2023, doi:10.3390/pharmaceutics15102368_

Round 1
Reviewer 1 Report
The study presents a detailed experimental investigation of the characteristics and performance of a delivery system combining a LFAC for dry powder surfactant aerosol with bCPAP for the treatment of premature infants. The work appears rigorous and sound. I have a few comments that require the authors' attention before recommending publication, as detailed below.
1) The quality of photographs in Figure 1 and Figure 6 shall be improved. It is very difficult to understand the elements shown.
2) In Figure 3 caption, when describing CFD analysis there are incorrect references to letters A, B and C.
3) In the description of the device, it is not clear how the LFAC bellows should be activated and synchronized with respect to the inhalation period. Is there any automatic means, i.e. by some controlled or pneumatically regulated device?
4) In general CFD results show great dependence on the boundary conditions. In Figure 3C the CFD analysis shows no flow from the bCPAP to the prong outlet. The timing of the results with respect to inhalation cycles should be reported. In addition, a proper CFD analysis should include transient evolution of the powder flow out of the capsule, dispersion in air and subsequent entrainment along the different prong geometries.
4) On page 8 there is a reference to Figure 4 which is unrelated to the text. It should be probably moved to Section "LFAC operational compatibility with bCPAP"
5) Discussion - Lung dose. The active humidified configuration is expected to generate cohesive-adhesive powder flow, with subsequent highly stable particle agglomeration and wall sticking. This is well known in industrial cohesive powder flow. The speculation on LFAC flow-induced possible de-agglomeration is not plausible.
Minor text corrections:
-) in Abstract - Study Design, correct "particle particle size"
-) double check the sentence "...the bellows to reinflate without entraining aerosol from the bellows" in Materials and Method and in Figure 1 caption.
-) Check the following sentence: "Each of the CPAP interfaces (n=5) were tested over six runs (n=5) for a total of 25 measurements".
-) Either use bCPAP or BCPAP or B-CPAP everywhere
-) note LAFC instead of LFAC in the "Safety and efficacy" section of the Discussion.
Author Response
We appreciate the reviewer’s careful critique and the many important suggestions regarding our manuscript.
Our responses, which we hope are satisfactory, are below.
Reviewer #1
1) The quality of photographs in Figure 1 and Figure 6 shall be improved. It is very difficult to understand the elements shown.
Response: Thank you, we replaced all of the figures with higher quality images.
2) In Figure 3 caption, when describing CFD analysis there are incorrect references to letters A, B and C.
Response: Thank you for noticing this error. We corrected it.
3) In the description of the device, it is not clear how the LFAC bellows should be activated and synchronized with respect to the inhalation period. Is there any automatic means, i.e. by some controlled or pneumatically regulated device?
Response: In the current study, the bellows were activated based on manual depression of the bellows and the visual observation of a chest rise analog of a pre-term infant. We added some more detail to clarify for readers. Some automated versions of this concept are still under development.
We added the following to Methods:
“An infant chest rise analog, consisting of a small silastic bladder and simulated infant chest wall, was intermittently inflated with an air compressor and timer that was turned on and off based on the flow signal (voltage) from the lung model. A single operator (RMD) was able to visualize chest rise and fall and manually depress the LFAC bellows to appropriately time (i.e. synchronize) LFAC actuation and aerosol delivery with lung model inhalation.”
And limitations:
A single operator manually activated the bellows based on the visual chest rise and fall using an infant chest rise analog. The delivery efficiency of DP surfactant aerosols may be heavily influenced by the operator’s ability to observe chest rise and appropriately time LFAC actuation with inhalation, especially in tachypneic newborns with severe RDS. Efforts should be made in the future to fully automate and synchronize inspiratory delivery of DP surfactant aerosols so that infants can receive more effective dosing.
4) In general CFD results show great dependence on the boundary conditions. In Figure 3C the CFD analysis shows no flow from the bCPAP to the prong outlet. The timing of the results with respect to inhalation cycles should be reported. In addition, a proper CFD analysis should include transient evolution of the powder flow out of the capsule, dispersion in air and subsequent entrainment along the different prong geometries.
Response: Thank you for this great input and for taking the time to observe these CFD effects. We especially appreciate the distinction you made based on the boundary condition which seems to isolate the CPAP flow. The boundary condition formed by the incoming LFAC flow is precisely what decouples the two flow sources and allows the aerosol to enter the prongs. We heavily edited the Figure 3 legend based on this feedback and revised/removed a lot of the speculation we had about CPAP gas flow mixing, which as you pointed out so elegantly, isn’t occurring. The timing for bellows activation, lung model flow and chest rise, and aerosol delivery was not perfectly synchronized and is likely to be slightly out of phase since actuation of bellows and aerosol generation was based on visual observation of chest rise and fall and not an electronic or pneumatic controller. We agree that all of these variables will need to be recorded in future studies to determine if there are any phase shifts that could potentially affect performance and delivery efficiency. The idea of CFD analysis on the entire system would be very useful in describing this highly complex system. We will be sure to include this in future publications.
Response:
5) On page 8 there is a reference to Figure 4 which is unrelated to the text. It should be probably moved to Section "LFAC operational compatibility with bCPAP"
Response: Thank you for noticing these errors. Now this moved to the result section of "LFAC operational compatibility with bCPAP."
6) Discussion - Lung dose. The active humidified configuration is expected to generate cohesive-adhesive powder flow, with subsequent highly stable particle agglomeration and wall sticking. This is well known in industrial cohesive powder flow. The speculation on LFAC flow-induced possible de-agglomeration is not plausible.
Response: We deleted this statement as suggested by the reviewer.
Minor text corrections:
-) in Abstract - Study Design, correct "particle particle size"
Response: Thank you for noticing this error. We corrected it.
-) double check the sentence "...the bellows to reinflate without entraining aerosol from the bellows" in Materials and Method and in Figure 1 caption.
Response: We corrected this sentence and revised to the following: “A series of one-way valves and a 3-way stopcock at the inlet of inhaler allows the bellows to reinflate from the atmosphere without entraining aerosol back into the LFAC capsule or inhaler chamber.”
-) Check the following sentence: "Each of the CPAP interfaces (n=5) were tested over six runs (n=5) for a total of 25 measurements".
Response: Corrected to total 30 measurements based on six runs for each of the five cannulae tested.
-) Either use bCPAP or BCPAP or B-CPAP everywhere
Response: We unified to bCPAP in this manuscript.
-) note LAFC instead of LFAC in the "Safety and efficacy" section of the Discussion.
Response: Corrected to LFAC.
Reviewer 2 Report
10.08.2023
A review to evaluate its suitability for publication Type of manuscript:
Article
Title: Evaluation of a Novel Dry Powder Surfactant Aerosol Delivery System for Use in Premature Infants Supported with Bubble CPAP
Authors: Robert M DiBlasi, Coral N Crandall, Rebecca J Engberg, Kunal Bijlani, Dolena R Ledee, Masaki Kajimoto, Frans J Walther
The relevance of this work is due to the need for mass distribution and introduction into pediatric practice of affordable devices for respiratory support and surfactant therapy of premature infants due to the low cost of equipment in low- and middle-income countries.
The aim of the work was to conduct preclinical in vitro studies to evaluate particle size, performance and aerosol delivery efficiency.
The authors' main findings were: no differences in nasal deposition and inhalation dose in the lungs between different nasal interfaces and gas moisture levels in the bCPAP system in simultaneous aerosol therapy with surfactant and bCPAP in the spontaneous breathing of a preterm infant.
The amount of work done by the authors is very impressive. The Introduction is excellently written, materials and methods of the study are described in great detail, including schemes and photographs of the equipment used.
The results of the work have important practical value, in general, the peer-reviewed manuscript is very important and relevant.
There are not many remarks to the work. The main remarks include: improving the quality of graphical drawings - Fig 4-7. They look very outdated with poorly legible data.
There is a need to check the text formatting requirements, e.g. figure captions.
The manuscript can be recommended for printing after responding to the comments of the Section Editor
Respectfully, reviewer
Author Response
We appreciate the reviewer’s careful critique and the many important suggestions regarding our manuscript.
Our responses, which we hope are satisfactory, are below.
Reviewer #2
1) There are not many remarks to the work. The main remarks include: improving the quality of graphical drawings - Fig 4-7. They look very outdated with poorly legible data.
Response: Thanks, you. We used color for these figures and increased the resolution which appeared to be lost when uploading the files initially.
2) There is a need to check the text formatting requirements, e.g. figure captions.
Response: Modified them.
Reviewer 3 Report
The manuscript presented by DiBlasi et al., showed the evaluation of a novel dry powder surfactant aerosol delivery system in premature infants supported with bubble CPAP. The dry powder was already evaluated in vivo (animal models) by the research group and is development was not discussed. The research was well conducted and scientifically correct.
Title. It could be changed. There is an emphasis for the “Use in premature Infants”. It generates an expectative that the DP was evaluated in infants in this study.
Abstract: Conclusions: please define the abbreviation NT.
Introduction. The authors may include in the 2nd or 3rd paragraph examples of the surfactants used.
Pg3. Introduction, last paragraph. Please specify that particle size was evaluated as MMAD. Also, the authors could shortly describe the development and the previous studies regarding this DP.
Results. Aerosol particle Size Distribution. Is it possible to include the standard deviation of the results?
Author Response
We appreciate the reviewer’s careful critique and the many important suggestions regarding our manuscript.
Our responses, which we hope are satisfactory, are below.
Reviewer #3
1) Title. It could be changed. There is an emphasis for the “Use in premature Infants”. It generates an expectative that the DP was evaluated in infants in this study.
Response: We prefer the title as is because our project aims at developing aerosol delivery of synthetic lung surfactant for “use” in premature infants.
2) Abstract: Conclusions: please define the abbreviation NT.
Response: Thank you for pointing out. NT is “nasotracheal,”. The CT scan of the nasal airway included the area from the nostrils to the nasopharynx and the airway model was connected to the lung model using a low dead space tracheal adaptor which together approximated the internal volume and resistance of the pre-term nasotracheal regions which we refer to simply as ‘nasal airways’ in this manuscript. So, we eliminated the use of the acronyms ‘NT’ and ‘NP’ in the manuscript to avoid confusion. Thank you for pointing this out.
3) Introduction. The authors may include in the 2nd or 3rd paragraph examples of the surfactants used.
Response: The development and previous studies regarding the various synthetic lung surfactants are included in the in the 2nd and 3rd paragraphs with references 12-16.
4) Pg3. Introduction, last paragraph. Please specify that particle size was evaluated as MMAD. Also, the authors could shortly describe the development and the previous studies regarding this DP.
Response: Great input. Thank You. Particle size as MMAD was inserted. We initially included a lot of citations that were based on the development of this DP surfactant, but the editor informed us that the author's self-citation rate should not exceed 15% of total references, so unfortunately, we had to remove many of them.
5) Results. Aerosol particle Size Distribution. Is it possible to include the standard deviation of the results?
Response: Unfortunately, standard statistics based on normal distributions are frequently not suitable for most airborne particle (aerosol) size distributions. In aerosol lognormal distributions, the log of the particle diameter is normally distributed. The MMAD is a statistically derived figure for a particle sample: for instance, an MMAD of 5 µm means that 50 % of the total sample mass will be present in particles having aerodynamic diameters less than 5 µm, and that 50 % of the total sample mass will be present in particles having an aerodynamic diameter larger than 5 µm. The normal linear based standard deviation with which most are familiar with is replaced with the standard deviation of the logarithms, called the geometric standard deviation (GSD). In the Results, we report the standard MMAD of 3.68 µm and GSD of 1.78. We were extremely limited in the abstract word count so we could not include or define all of these acronyms, so we reported the ‘particle size’ as a general MMAD value.